# Simulation and Investigation of the Change of Geometric Parameters on Voltage Induced in the Energy Harvesting System with Magnetic Spring

Joanna Bijak *, Tomasz Trawiński and Marcin Szczygieł

Faculty of Electrical Engineering, Department of Mechatronics, Silesian University of Technology,
44-100 Gliwice, Poland; tomasz.trawinski@polsl.pl (T.T.); marcin.szczygiel@polsl.pl (M.S.)
* Correspondence: joanna.bijak@polsl.pl

**Abstract:** The aim of this paper is to establish mathematical modelling and simulation for the voltage induced during movement of the moveable magnet in a double-sided magnetic spring, being part of the energy harvesting system. For various configurations of the set of permanent magnets, the repulsive forces of magnetic spring and induced voltage in energy harvester winding will be calculated. Changing the geometrical dimensions and shape of permanent magnets allows one to control the stiffness of the so-called double-sided magnetic spring, and furthermore, allows one to change the natural frequency of the energy harvester system. Properly chosen stiffness in the energy harvester system is the crucial issue for high efficiency in energy recovery. In a given case, the energy harvester consists of three permanent magnets inserted into a tube with coils wound on it. To calculate the force between the magnets and the magnetic flux in the coils, the ANSYS program was used. The voltages induced in coils for various configurations of the magnets were simulated in the MATLAB program.

**Keywords:** energy harvesting; energy harvester; magnetic spring; mechanical vibration

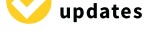



## 1. Introduction

Nowadays, energy is harvested from as many sources as possible, so as not only to reduce pollution and damage from the extraction of non-renewable energy sources but also for reducing the system usage cost. The energy harvesting system allows one to provide wireless power supply in hard-to-reach places, such as on bridges, to monitor their stress [1], and in mobile systems, such as in cars, for tire pressure monitoring systems (TPMS) [2], or to supply actives dampers [3] or provide additional supply to batteries, such as in mobile phones, where the energy is harvested from movement of the human body [4]. Mechanical vibration energy harvesters can be piezoelectric, electrostatic or electromagnetic generators [5]. The cheapest cost of production is for electromagnetic generators and, therefore, they are the most popular [6]. Electromagnetic generators can harvest high power from low-frequency vibrations; low-frequency vibration is common for most household appliances and vehicles [7,8]. An electromagnetic energy harvesting system, which uses a movable magnet in a coil, can use a magnetic spring to allow magnet movement in a range limit set by the external magnets. The relative movement of a magnet and coil induces electromotive force. The electromotive force depends on the following parameters: the relative velocity of a magnet and coil, the magnetic flux density distribution in coil volume, radius of a coil, and a coil area. To maximize the energy harvested in that system, it is possible to change the parameters mentioned earlier. In order to enhance the magnetic flux, it is possible to replace a single magnet with two or more. With some material with different permeability between them [9–11], magnets can be positioned in opposing or matching orientations to each other [12]. To adjust the resonance frequency, it is possible to change magnet parameters, such as diameter, height, mass [9,10,13,14],

shape [13], quantity [9,10] and the distance between them. Sometimes, that magnet can be positioned by the additional mechanical springs or guiding rods/rolls [8,9,15,16]. The overview of energy harvesters with various magnetic springs (with multiple coils, with one coil, with one moveable magnet or with multiple moveable magnets) is described in [17]. Magnetic springs have been examined by various authors and there are many possible configurations, but parameters for the magnetic springs and whole energy harvester vary too much to compare them [17].

The huge disadvantage of all the generators is that they usually need to work in the resonance frequency in order to provide significant power [5]. The resonance frequency in several systems, such as in the mentioned car dampers or TPMS, or during the human movement or vibration of the bridges, have various resonant frequency, so the best generator should be one with variable resonance frequency [5]. One of such a solution could be a magnetic spring, because the resonance frequency of such a spring relies on the magnet parameters. For some of the mentioned systems, these parameters can be changed during vibration energy harvesting. A magnetic spring with variable stiffness was shown in [18], where the stiffness was changed by the two ring-shaped magnets and the relative position between them. In [19], variable resonance frequency was obtained by magnets placed on a beam at different distances. In [12], the resonance frequency of the magnetic spring (the stiffness of the magnetic spring) was changed by changing the gap between the magnets. In [20], the resonance frequency was lowered by the negative stiffness of the magnetic spring and the differences in the magnetic force were investigated for the different radius and height of the fixed magnet; the internal magnet was ring-shape. In [21], the resonance frequency was adjusted by changing the air gap between two magnets, which was the result of the coil excitation change. In [22], the authors investigated the change in the linear and nonlinear stiffness coefficient of the spring in comparison to the change of such parameters as heights and diameters of the fixed magnets, the distance between them and the height and diameter of the levitated magnet. However, they did not answer the question of which way to adjust resonance frequency without changing the power significantly. Optimization of the springs was made by several authors but they focused on the power maximization for given force and on the magnetic spring model for given frequency, for example in [23] and [24], not on the best way to adjust stiffness. Therefore, the goal of this article was to investigate the influence of the magnet parameter change on the stiffness of the magnetic spring. After that, the maximum power for each magnetic spring configuration, while the energy harvester is working in resonance frequency, is compared. As a result, the best and easiest way to match resonance frequency can be chosen.

In this paper, a finite element and mathematical model for a double-sided magnetic spring with two coils wound on it, which is acting as an electromagnetic energy harvesting system (EHS), is considered for simulation of the parameters of the magnetic spring and the voltage induced during movement of the magnet. A double-sided magnetic spring consists of two fixed magnets (external magnets) and a magnet suspended between them (internal magnet); a magnet is moving due to vibrations and its movement is limited by fixed magnets and a tube surrounding the whole. The repulsive force between an external magnet and an internal magnet can be considered as the spring force; therefore, these sets of magnets are often called a magnetic spring. The discussed double-sided magnetic spring consists of two such springs connected by internal magnets. External magnets have, in that case, the same parameters, so two magnetic springs have the same stiffness.

Various configurations of the magnetic spring, such as magnet width, height, and shape (either cylindrical or ring magnet), and the voltage induced for such sets of configurations, were investigated. These parameters are described in Section 2 of this paper. The forces of the springs change with these parameters and they were investigated using FEM (finite element method) in the ANSYS program. The forces and stiffnesses, which are differential from the force by displacement of the magnet, were compared in Section 3 of this paper. Movement of the magnet in the magnetic spring was obtained from the mathematical model described by a set of ordinary differential equations with lumped

parameters, describing the motion of mass system suspended by springs, called the OE model. The model formulated with the help of the FEM (finite element method) and OE model can be used to determine parameters in the magnetic spring to adjust the resonant frequency of the energy harvester to reach possibly higher harvested energy. In the ANSYS Electronics program, it is possible to simulate the magnetic flux in a coil with specified parameters. Therefore, the voltage induced in a coil can be calculated as a time differential of the magnetic flux. The model for the movement calculation and comparison of voltage induced in coils and power are shown in Section 4. The best and easiest way to change the resonance frequency was chosen in the conclusions in Section 5.

## 2. Energy Harvester with Magnetic Spring

In the case where the magnetic flux can be determined, as in the case of our simulation, the equation of the voltage $e$ induced in a coil can be calculated directly from the Faraday law:

$$e = -\frac{\mathrm{d}\phi}{\mathrm{dt}}, \tag{1}$$

where $\phi$—the magnetic flux.

The magnetic flux depends on the position of a magnet, so (1) can be substituted using partial differentials:

$$e = -\frac{\partial\phi}{\partial z}\frac{\partial z}{\partial t}, \tag{2}$$

where $z$—the position of the magnet.

In Equation (2), it can be seen that the induced voltage depends on the position of a magnet and its velocity. From the system point of view, the electromagnetic energy harvester consists of two systems, which have an influence on each other: electromagnetic and mechanical. The displacement and resulting velocity of a permanent magnet inside the coil from the mechanical system are input data for the electromagnetic system and allows one to produce the voltage. In the electromagnetic system, the magnetic field of magnets has an influence on the magnetic spring coefficient; the voltage induced in the coil causes the appearance of the electromagnetic force, which acts as a damping force in the mechanical system, when the harvested energy is consumed or stored [25,26].

The repulsive force between the magnets can be considered as a non-linear spring force, which was modeled using finite element methods (FEM) in the ANSYS Electronics program. This force has an influence on the resonant frequency through the stiffness coefficient. That can be changed by altering the parameters of the double-sided magnetic spring. In the model, the diameter and height of the magnets were altered, as well as the air gap between magnets; magnets were either cylindrical or ring-shaped. The influence of changes on the repulsive force acting on the internal magnet and voltage induced in winding (coils) was investigated. The winding is cylindrical with a longitudinal axis along the magnet's axis of the symmetry. It consists of two identical coils connected in series. Coil parameters depend on the double-sided magnetic spring dimensions. The coils were placed symmetrically on both sides of the double-sided magnetic spring. The coils' internal radiuses $r_{ci}$ were equal to the internal magnet radius plus 1 mm, when the diameter of the internal magnet is greater than the diameter of an external magnet or 1 mm plus radius of the external magnet if it is the other way. The external radiuses of the coils $r_{ce}$ were 3 mm bigger than the internal radius. The heights of the coils $h_c$ were equal to the distance between the external magnets minus 1 mm and divided by 2. Coils were placed on the height $h_{cs}$, meaning 0.5 mm above and below the center of a double-sided magnetic spring. The number of coil turns was determined by the coil dimensions. For the coil with height of 5 mm and width equal to 3 mm, the number of turns $N$ equals 400. For a higher coil, this number (400) was multiplied by the height of a recent coil divided by 5 mm.

Simulations were performed for the variable position of an internal magnet position in optimetrics, set in the Ansys Electronics program, with the number of steps depending on the air gap; the value of the position change was 0.01 mm.

For the analysis, a few different cases of dimensions (shapes: cylindrical or ring-shaped) in the magnets and dimensions between magnets were chosen. The permanent magnets were neodymium magnets (NdFeB) N38, with parameters given in [27]. The dimensions were altered to maintain constant proportions between some of them. The dimensions that were altered are shown in Figure 1a. The winding dimensions are shown in Figure 1b.

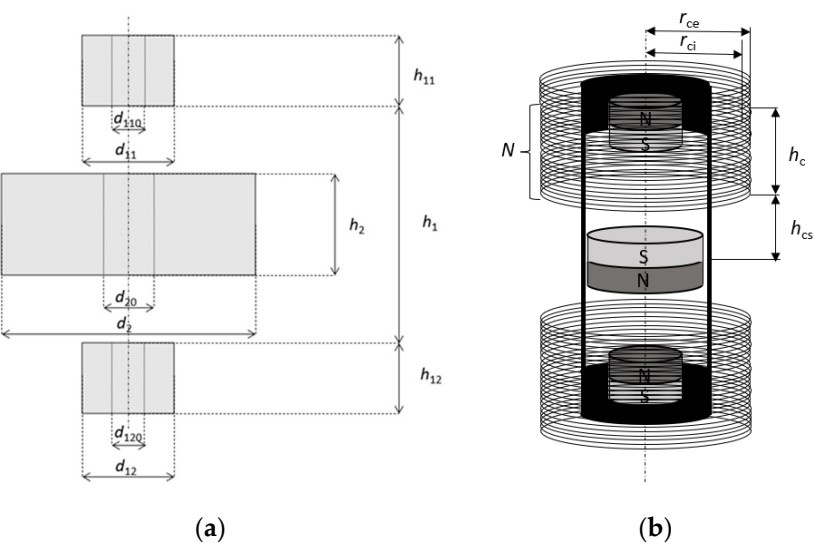

|     | (a) | (b) |

**Figure 1.** Dimensions of (**a**) the magnetic spring, (**b**) windings.

The sets of cases are described as functions $s_i c_n([h_1, h_2, h_2/h_1], [h_{11}, h_{12}, h_{11}/h_2, h_{12}/h_2], [d_{11}, d_{12}, d_2, d_2/d_{11}, d_2/d_{12}], [d_{110}, d_{120}, d_{20}, d_{110}/d_{11}, d_{120}/d_{12}], [h_2/d_2, h_{11}/d_{11}, h_{12}/d_{12}])$, in which dimensions are as marked in Figure 1, where $i$ is the number of a set and $n$ is number of a case. If dimensions $d_{110}, d_{120}, d_{20}$ equal 0, there is no hole in the permanent magnets and the magnets are cylindrical. In all cases, two external magnets have the same parameters; therefore, $h_{11} = h_{12}$, $d_{11} = d_{12}$ and $d_{110} = d_{120}$. Sets of cases for double-sided magnetic springs are shown in tables, with pictograms illustrating changes in shapes for the cases in the right column. The sets of cases for cylindrical magnets are shown in Table 1, for ring-shaped magnets in Table 2, and for proportional cases in Table 3. The set number in the tables is denoted by $s$, case number in the tables is denoted by $c$, $h$ are heights and $d$ diameters of magnets in millimeters. *Ref* and *ref0* are reference cases to which all cases are compared. *Ref* is the reference case for cylindrical magnets and *ref0* is the reference case for ring-shaped magnets. The reference double-sided magnetic spring to which all cases are compared has cylindrical magnets and the following parameters: $h_1 = 11$, $h_2 = 3$, $h_{11} = h_{12} = 5$, $d_{11} = d_{12} = 5$, $d_2 = 10$. The reference magnetic spring was the real object whose behavior on the vibration generator was investigated. The rest of the cases were chosen adequately for this magnetic spring, keeping several proportions, compared to the reference magnetic spring. The dimensions were multiplied by 2 and 3 to allow for the investigation of such springs in future works. Dimensions of the ring-shaped magnet were chosen from 1 to 3 mm in order to facilitate the production of such a magnet. In order to check which parameter has the greatest impact on the change in the stiffness after the two first cases, the third case of the magnetic spring was added (changing of all magnet heights).

**Table 1.** Set of cases with cylindrical magnets.

| $s$ | $c$ | $h$, mm | $d$, mm | Pictogram |
|---|---|---|---|---|
| *Ref* | *Ref* | $h_{11} = 5$<br>$h_2 = 3$<br>$h_1 = 11$ | $d_{11} = 5$<br>$d_2 = 10$<br>$d_{110} = 0$ | |
| 1 | 2 | $h_{11} = 5$<br>$h_2 = 6$<br><br>$h_1 = 22$ | $d_{11} = 5$<br>$d_2 = 10$<br><br>$d_{110} = 0$ | |
| 1 | 3 | $h_{11} = 5$<br>$h_2 = 9$<br><br>$h_1 = 33$ | $d_{11} = 5$<br>$d_2 = 10$<br><br>$d_{110} = 0$ | |
| 2 | 2 | $h_{11} = 10$<br>$h_2 = 6$<br><br>$h_1 = 11$ | $d_{11} = 5$<br>$d_2 = 10$<br><br>$d_{110} = 0$ | |
| 2 | 3 | $h_{11} = 15$<br>$h_2 = 9$<br><br>$h_1 = 11$ | $d_{11} = 5$<br>$d_2 = 10$<br><br>$d_{110} = 0$ | |
| 3 | 2 | $h_{11} = 10$<br>$h_2 = 6$<br><br>$h_1 = 22$ | $d_{11} = 5$<br>$d_2 = 10$<br><br>$d_{110} = 0$ | |
| 3 | 3 | $h_{11} = 15$<br><br>$h_2 = 9$<br><br>$h_1 = 33$ | $d_{11} = 5$<br><br>$d_2 = 10$<br><br>$d_{110} = 0$ | |
| 4 | 2 | $h_{11} = 5$<br>$h_2 = 3$<br>$h_1 = 11$ | $d_{11} = 10$<br>$d_2 = 20$<br>$d_{110} = 0$ | |
| 4 | 3 | $h_{11} = 5$<br>$h_2 = 3$<br>$h_1 = 11$ | $d_{11} = 15$<br>$d_2 = 30$<br>$d_{110} = 0$ | |

**Table 2.** Set of cases with ring-shaped magnets.

| $s$ | $c$ | $h$, mm | $d$, mm | Pictogram |
|---|---|---|---|---|
| *ref0* | *ref0* | $h_{11} = 5$<br>$h_2 = 3$<br>$h_1 = 11$ | $d_{11} = 5$<br>$d_2 = 10$<br>$d_{110} = 3$ | |
| 5 | 2 | $h_{11} = 5$<br>$h_2 = 3$<br>$h_1 = 11$ | $d_{11} = 5$<br>$d_2 = 10$<br>$d_{110} = 1$ | |
| 5 | 3 | $h_{11} = 5$<br>$h_2 = 3$<br>$h_1 = 11$ | $d_{11} = 5$<br>$d_2 = 10$<br>$d_{110} = 2$ | |
| 6 | 2 | $h_{11} = 5$<br>$h_2 = 3$<br>$h_1 = 11$ | $d_{11} = 10$<br>$d_2 = 10$<br>$d_{110} = 6$ | |
| 6 | 3 | $h_{11} = 5$<br>$h_2 = 3$<br>$h_1 = 11$ | $d_{11} = 15$<br>$d_2 = 10$<br>$d_{110} = 9$ | |

**Table 3.** Set of cases for proportional magnetic springs.

| $s$ | $c$ | $h$, mm | $d$, mm | Pictogram |
|---|---|---|---|---|
| *Ref* | *Ref* | $h_{11} = 5$<br>$h_2 = 3$<br>$h_1 = 11$ | $d_{11} = 5$<br>$d_2 = 10$<br>$d_{110} = 0$ | |
| 7 | 2 | $h_{11} = 10$<br>$h_2 = 6$<br>$h_1 = 22$ | $d_{11} = 10$<br>$d_2 = 20$<br>$d_{110} = 0$ | |
| 7 | 3 | $h_{11} = 15$<br>$h_2 = 9$<br>$h_1 = 33$ | $d_{11} = 15$<br>$d_2 = 30$<br>$d_{110} = 0$ | |
| *ref0* | *ref0* | $h_{11} = 5$<br>$h_2 = 3$<br>$h_1 = 11$ | $d_{11} = 5$<br>$d_2 = 10$<br>$d_{110} = 3$ | |
| 8 | 2 | $h_{11} = 10$<br>$h_2 = 6$<br>$h_1 = 22$ | $d_{11} = 10$<br>$d_2 = 20$<br>$d_{110} = 6$ | |
| 8 | 3 | $h_{11} = 15$<br>$h_2 = 9$<br>$h_1 = 33$ | $d_{11} = 15$<br>$d_2 = 30$<br>$d_{110} = 9$ | |

## 3. Repulsive Forces and Spring Stiffness

The repulsive force acting on the moveable magnet from the fixed magnets was calculated as the virtual force from the differentiation of energy, as follows:

$$F_{\text{k}}(i) = \frac{W_{\text{e}}(i) - W_{\text{e}}(i+1)}{\delta z},\tag{3}$$

where $F_k(i)$—repulsive force calculated in $i$ step, $W_e(i)$—total energy of the moveable magnet area calculated in $i$ step, $W_e(i + 1)$—total energy of the moveable magnet area calculated in $i + 1$ step, $\delta z$—displacement of the moveable magnet (from step $i$ to $i + 1$) (0.1 mm).

After FEM calculations, the characteristics of forces were approximated using 9-degree polynomial, so the force can be presented as follows:

$$F_k(z) = a_1 z^9 + a_2 z^8 + \ldots + a_8 z^2 + a_9 z + a_0, \tag{4}$$

where $a_i \in \{0, 1, 2, \ldots, 8, 9\}$—coefficients of the approximation polynomial of the 9th degree. The coefficient $a_0$ can be omitted. These coefficients for all cases are shown in Table 4.

**Table 4.** Coefficients of the approximation polynomial of the force.

| $s$ | $c$ | $a_1$ | $a_2$ | $a_3$ | $a_4$ | $a_5$ | $a_6$ | $a_7$ | $a_8$ | $a_9$ |
|---|---|---|---|---|---|---|---|---|---|---|
| *Ref* | *Ref* | $-7.34 \times 10^{21}$ | $-3.16 \times 10^{17}$ | $1.61 \times 10^{17}$ | $6.66 \times 10^{12}$ | $-1.88 \times 10^{12}$ | $-2.66 \times 10^{7}$ | $1.91 \times 10^{7}$ | $60.24$ | $958.72$ |
| 1 | 2 | $-4.14 \times 10^{19}$ | $7.85 \times 10^{15}$ | $3.53 \times 10^{15}$ | $-7.43 \times 10^{11}$ | $-7.46 \times 10^{10}$ | $1.84 \times 10^{7}$ | $7.94 \times 10^{6}$ | $-91.85$ | $338.72$ |
| 1 | 3 | $-2.51 \times 10^{18}$ | $1.98 \times 10^{14}$ | $6.21 \times 10^{14}$ | $-4.23 \times 10^{10}$ | $-3.46 \times 10^{10}$ | $2.48 \times 10^{6}$ | $2.87 \times 10^{6}$ | $-52.57$ | $119.62$ |
| 2 | 2 | $-5.31 \times 10^{23}$ | $1.60 \times 10^{20}$ | $4.67 \times 10^{18}$ | $-1.80 \times 10^{15}$ | $-1.48 \times 10^{13}$ | $6.26 \times 10^{9}$ | $1.22 \times 10^{7}$ | $-6.76 \times 10^{3}$ | $2.51 \times 10^{3}$ |
| 2 | 3 | $-9.35 \times 10^{26}$ | $2.94 \times 10^{23}$ | $1.42 \times 10^{21}$ | $-4.10 \times 10^{17}$ | $-7.18 \times 10^{14}$ | $1.49 \times 10^{11}$ | $-1.11 \times 10^{8}$ | $-1.30 \times 10^{4}$ | $4.00 \times 10^{3}$ |
| 3 | 2 | $-3.67 \times 10^{19}$ | $-1.19 \times 10^{16}$ | $3.04 \times 10^{15}$ | $1.24 \times 10^{12}$ | $-5.00 \times 10^{10}$ | $-3.75 \times 10^{7}$ | $8.82 \times 10^{6}$ | $303.05$ | $421.64$ |
| 3 | 3 | $-2.24 \times 10^{18}$ | $1.56 \times 10^{14}$ | $5.62 \times 10^{14}$ | $-2.03 \times 10^{10}$ | $-2.76 \times 10^{10}$ | $-1.41 \times 10^{5}$ | $3.21 \times 10^{6}$ | $42.32$ | $172.88$ |
| 4 | 2 | $-1.14 \times 10^{22}$ | $5.31 \times 10^{18}$ | $3.46 \times 10^{17}$ | $-1.61 \times 10^{14}$ | $-4.48 \times 10^{12}$ | $1.48 \times 10^{9}$ | $-3.62 \times 10^{6}$ | $-3.91 \times 10^{3}$ | $1.30 \times 10^{3}$ |
| 4 | 3 | $-2.76 \times 10^{22}$ | $-5.46 \times 10^{18}$ | $8.48 \times 10^{17}$ | $1.43 \times 10^{14}$ | $-9.38 \times 10^{12}$ | $-1.03 \times 10^{9}$ | $8.33 \times 10^{6}$ | $1.54 \times 10^{3}$ | $600.65$ |
| *ref0* | *ref0* | $-5.84 \times 10^{21}$ | $5.59 \times 10^{17}$ | $9.08 \times 10^{16}$ | $-1.09 \times 10^{13}$ | $-4.69 \times 10^{11}$ | $7.11 \times 10^{7}$ | $1.05 \times 10^{7}$ | $-433.71$ | $608.81$ |
| 5 | 2 | $-4.64 \times 10^{21}$ | $4.38 \times 10^{17}$ | $5.47 \times 10^{16}$ | $-1.07 \times 10^{13}$ | $-3.32 \times 10^{11}$ | $2.80 \times 10^{7}$ | $1.19 \times 10^{7}$ | $469.85$ | $805.51$ |
| 5 | 3 | $-4.24 \times 10^{21}$ | $-2.35 \times 10^{17}$ | $5.48 \times 10^{16}$ | $9.59 \times 10^{12}$ | $-5.16 \times 10^{11}$ | $-1.32 \times 10^{8}$ | $1.29 \times 10^{7}$ | $615.98$ | $921.08$ |
| 6 | 2 | $3.00 \times 10^{22}$ | $-7.88 \times 10^{18}$ | $-6.46 \times 10^{17}$ | $1.98 \times 10^{14}$ | $8.82 \times 10^{12}$ | $-1.48 \times 10^{9}$ | $3.20 \times 10^{7}$ | $3.33 \times 10^{3}$ | $1.43 \times 10^{3}$ |
| 6 | 3 | $-2.13 \times 10^{22}$ | $-7.83 \times 10^{18}$ | $4.11 \times 10^{17}$ | $2.01 \times 10^{14}$ | $-6.58 \times 10^{12}$ | $-1.56 \times 10^{9}$ | $-4.41 \times 10^{7}$ | $3.90 \times 10^{3}$ | $129.01$ |
| 7 | 2 | $-2.02 \times 10^{19}$ | $-2.04 \times 10^{16}$ | $8.23 \times 10^{14}$ | $2.80 \times 10^{12}$ | $-1.01 \times 10^{11}$ | $-1.20 \times 10^{8}$ | $8.58 \times 10^{6}$ | $1.67 \times 10^{3}$ | $1.92 \times 10^{3}$ |
| 7 | 3 | $6.40 \times 10^{17}$ | $1.87 \times 10^{15}$ | $-5.53 \times 10^{14}$ | $-4.97 \times 10^{11}$ | $5.37 \times 10^{10}$ | $4.51 \times 10^{7}$ | $1.63 \times 10^{6}$ | $-1.53 \times 10^{3}$ | $2.93 \times 10^{3}$ |
| 8 | 2 | $-8.84 \times 10^{19}$ | $1.20 \times 10^{16}$ | $9.11 \times 10^{15}$ | $-1.20 \times 10^{12}$ | $-3.91 \times 10^{11}$ | $3.79 \times 10^{7}$ | $1.24 \times 10^{7}$ | $-431.01$ | $1.18 \times 10^{3}$ |
| 8 | 3 | $-4.33 \times 10^{18}$ | $3.29 \times 10^{14}$ | $8.08 \times 10^{14}$ | $-7.69 \times 10^{10}$ | $-5.63 \times 10^{10}$ | $6.07 \times 10^{6}$ | $4.81 \times 10^{6}$ | $-269.25$ | $1.82 \times 10^{3}$ |

The repulsive forces acting on the moveable (internal) magnet are shown in Figures 2–4.

In Figure 2a, the set of curves representing the repulsive force acting on the movable magnets, with the geometric configuration belonging to set $s_1$, where $h_2/h_1 = $ const. for $h_2 = 3$ and $h_1 = 11$, $h_2 = 6$ and $h_1 = 22$, $h_2 = 9$ and $h_1 = 33$, are presented. The higher the distance between external magnets, the more non-linear the curves are. Compared to other curves, the force curve for the reference double-sided magnetic spring is almost linear. The maximum repulsive force increases with an increase in the height of the internal magnet and the distance between external magnets. However, in the center, so in the range of the magnet's expected movement, the force is lower for the higher internal magnets and the distance between external magnets is higher. The slope of the curve in this range is lower for the higher internal magnet and the higher distance between external magnets, so the stiffness will be lower. In Figure 2b, it can be seen that the higher the internal and external magnets, the lower the slope of the curve. In Figure 2c, it can be seen that the higher the magnets are and the longer the distance between them is, the higher the maximum force acting on the internal magnet is. However, in the internal magnet movement range, the force is lower for higher magnets and distance between them. The slope of the linear part of the curve is higher than in Figure 2a. Comparing Figure 2a–c, it can be seen that the linearity of the curve depends on the air gap between magnets; the longer the distance between them, the more nonlinear the curve is. The higher internal magnet, in comparison to the air gap, causes a higher slope in the linear part of the curve, which is consistent

with [12], where for the higher air gap, the force was lower. Therefore, the higher the proportions between the internal magnet and the air gap, the higher the stiffness in the magnetic spring is. The higher the external magnets and the internal magnet, the higher the force acting on the internal magnet is. The higher magnets generally increase the slope of the linear part of the curve. In Figure 2d, it can be seen that the higher the diameter of the magnets, the lower the force and the more non-linear the curve is.

In Figure 3, we can see the set of curves that represent the repulsive force acting between the ring-shaped magnets. The geometric configuration belonging to set $s_5$, where the internal diameter of the ring-shaped external magnet is changing, is presented in Figure 3a. It can be seen that the higher value for the diameter causes a decrease in the maximum value of the force and the slope of the linear part of the curve. In Figure 3b, the geometric configuration belonging to set $s_6$ is shown, and the internal and external diameters of the external magnet change proportionally. The higher the diameters, the higher the slope of the linear part of the curve is, and the more nonlinear the curve is. It can be seen that, when the external magnet has the highest diameter, the force direction changes.

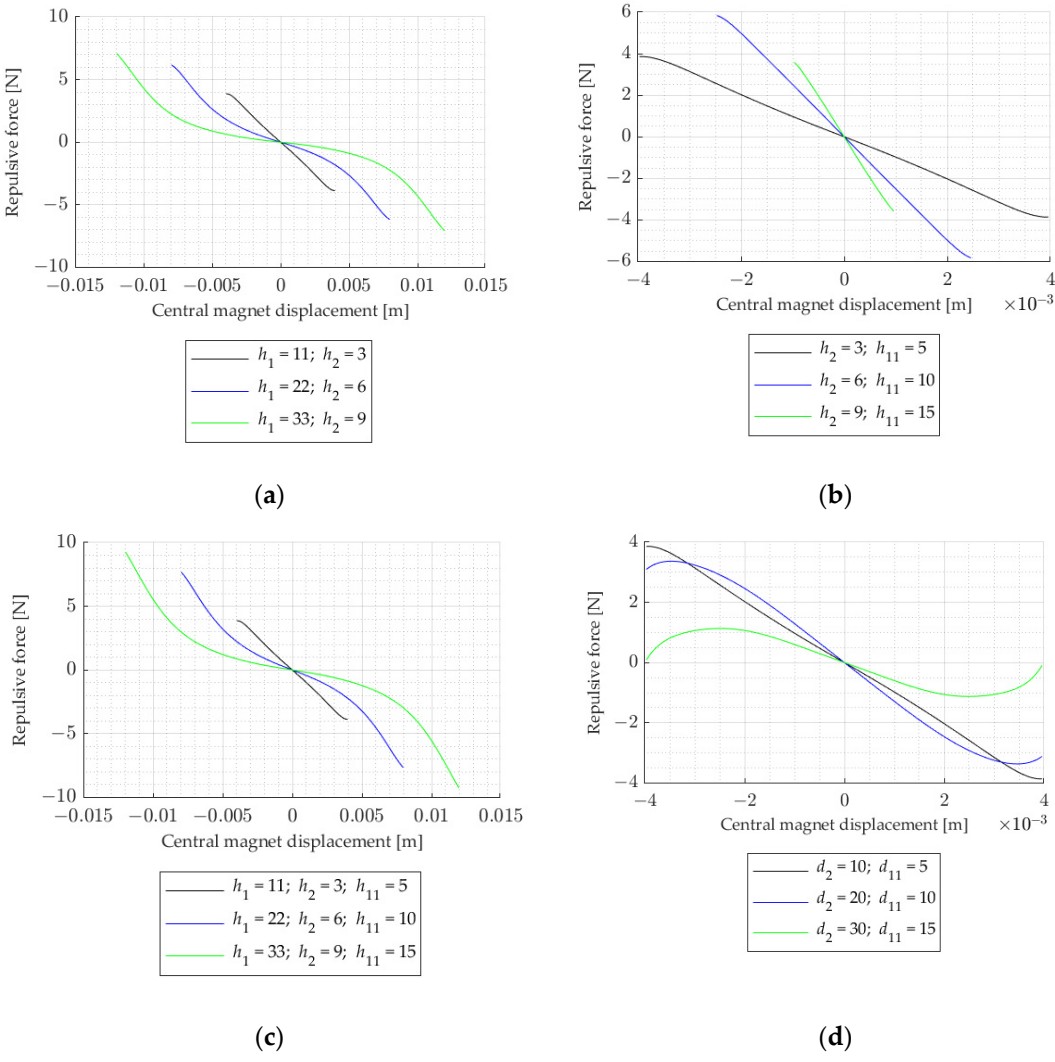

**Figure 2.** The repulsive force acting on a central magnet: (**a**) set $s_1$ $h_2/h_1$ = const. (**b**) set $s_2$ $h_{11}/h_2$ = $h_{12}/h_2$ = const. (**c**) set $s_3$ $h_2/h_1$ = const. and $h_{11}/h_2$ = $h_{12}/h_2$ = const. (**d**) set $s_4$ $d_{11}/d_2$ = $d_{12}/d_2$ = const.

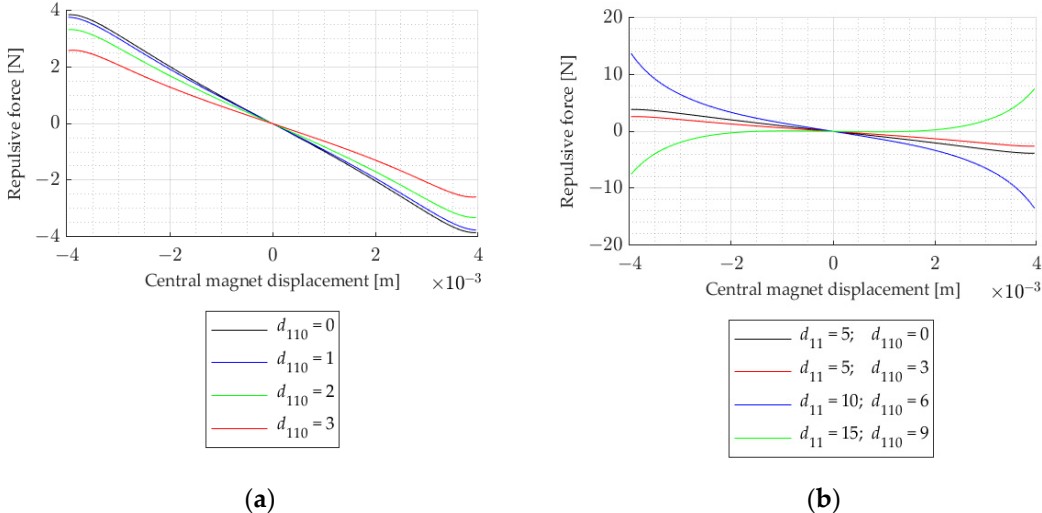

(**a**)    (**b**)

**Figure 3.** The repulsive force acting on a central magnet: (**a**) set $s_5$ various internal diameter of the external magnets, (**b**) set $s_6$ $d_{110}/d_{11} = d_{120}/d_{12} = const$.

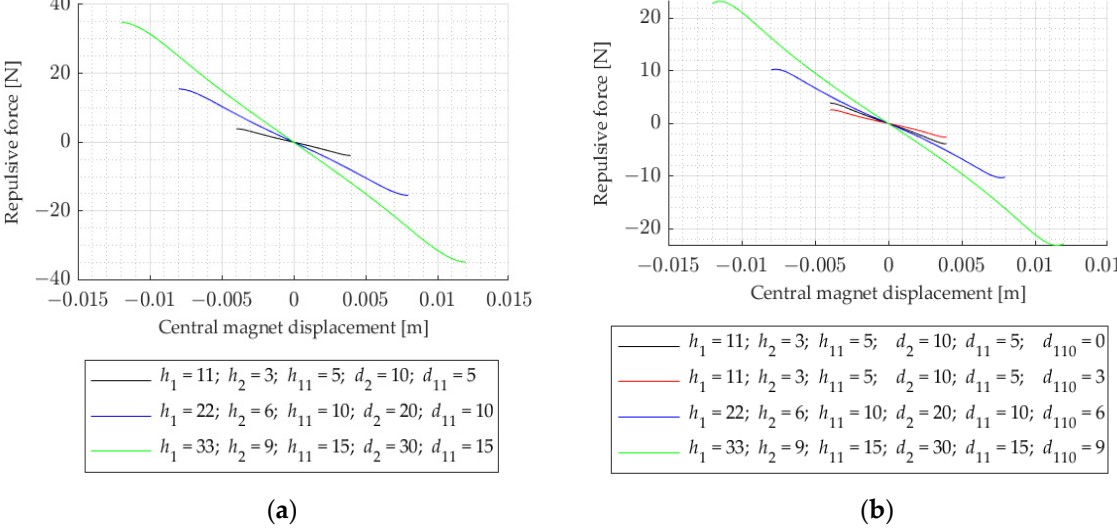

(**a**)    (**b**)

**Figure 4.** The repulsive force acting on a central magnet: (**a**) set $s_7$ with cylindrical magnets, where all proportions are kept constant, and (**b**) set $s_8$ with ring-shaped magnets, where all proportions are kept constant.

In Figure 4a, the geometric configuration belonging to set $s_7$ is shown and in Figure 4b, the geometric configuration belonging to set $s_8$ is shown. All the geometric parameters of the magnets change proportionally. It can be seen that the higher the values of these parameters, the higher the force and the higher the slope of the linear part of the curve is.

The stiffness of the spring can be calculated as a derivative by $z$ from the force (4) and is shown in Figures 5–7.

Comparing the curves in Figure 5a–c, it can be seen that the higher the distance between external magnets, the smaller the stiffness coefficient is. The stiffness curve shapes are closer to parabolic shapes. For higher magnets, the stiffness coefficient is higher, which is consistent with [22]. Lowering the gap between magnets by changing internal magnet dimensions, without changing the distance between external magnets, causes the linearization of the curve, but if the internal magnet is too close to the external magnets, the curve is similar to the negative parabolic curve. When the internal magnet is near that of the external magnets, the influence of that external magnet is higher than the influence of

the second external magnet. The closer magnets are to each other, the more significant the attraction force becomes. The lower the height of the middle magnet, the sooner it happens, meaning for higher distances between the middle and external magnets, the middle magnet will vibrate or it will rotate if it does not have any guiding rod. In Figure 5d, it can be seen that by increasing the diameters of the magnets, the difference between the maximum and minimum value in the stiffness coefficient curve increases and the minimum value in the stiffness coefficient curve decreases. In a range ±1 mm, for higher diameters, the curve is horizontal and linear.

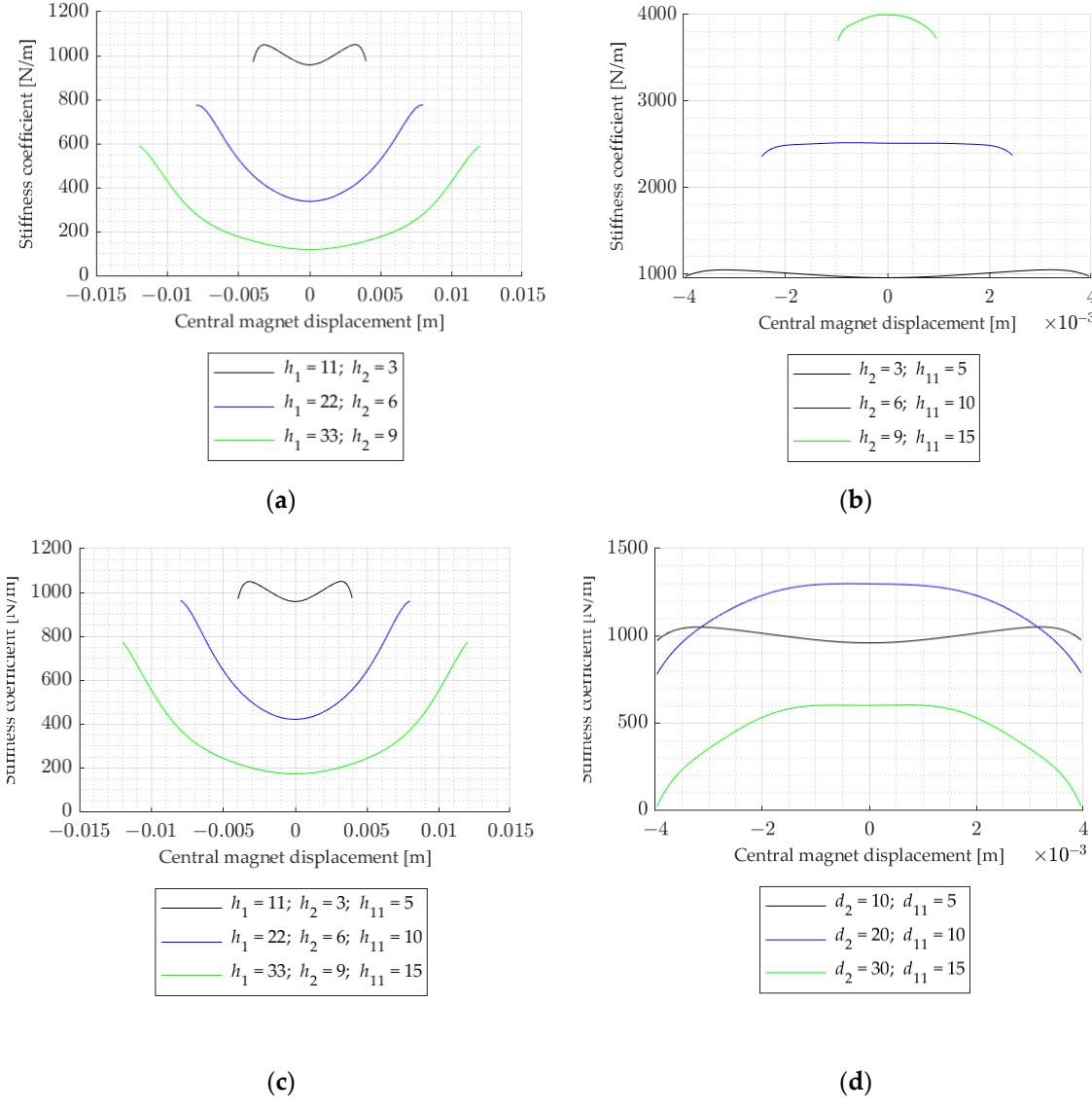

**Figure 5.** The double-sided magnetic spring stiffness coefficient: (**a**) set $s_1$ $h_2/h_1$ = const. (**b**) set $s_2$ $h_{11}/h_2 = h_{12}/h_2$ = const. (**c**) set $s_3$ $h_2/h_1$ = const. and $h_{11}/h_2 = h_{12}/h_2$ = const. (**d**) set $s_4$ $d_{11}/d_2 = d_{12}/d_2$ = const.

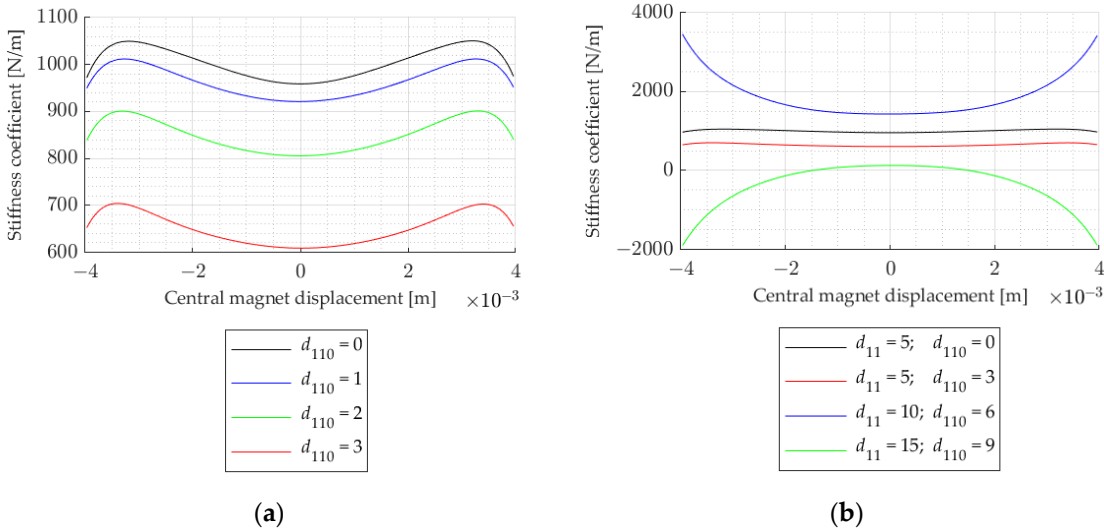

(**a**)                                              (**b**)

**Figure 6.** The double-sided magnetic spring stiffness coefficient: (**a**) set $s_5$ various internal diameters of the external magnets, (**b**) set $s_6$ $d_{110}/d_{11} = d_{120}/d_{12}$ = const.

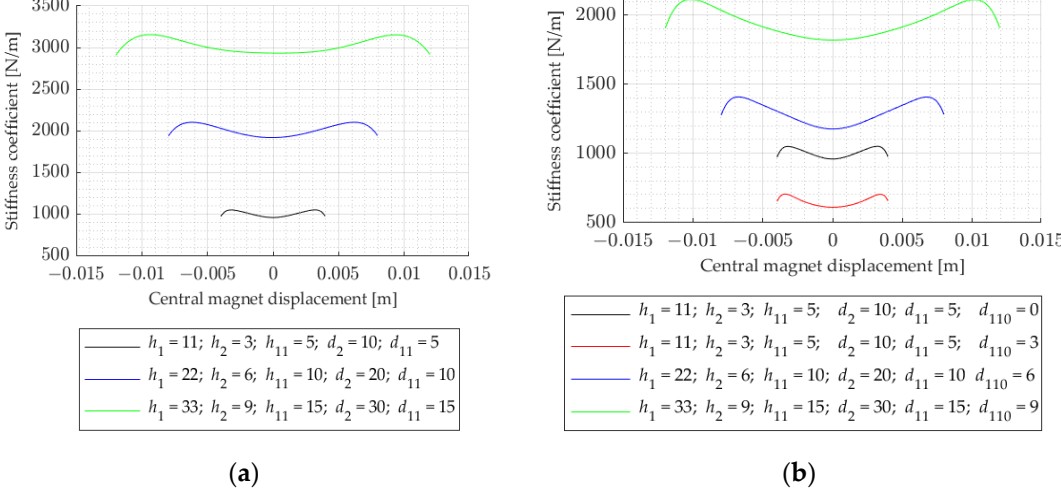

(**a**)                                              (**b**)

**Figure 7.** The double-sided magnetic spring stiffness coefficient: (**a**) set $s_7$ with cylindrical magnets, where all proportions are kept constant, and (**b**) set $s_8$ with ring-shaped magnets, where all proportions are kept constant.

In Figure 6a, the set of curves representing the stiffness coefficient for the double-sided magnetic spring with the geometric configuration belonging to set $s_5$ is presented. The higher the diameter value, the lower the stiffness coefficient (consistent with [22]) and the higher the slope of the curve is, when the internal magnet is near external magnets. In Figure 6b, it can be seen that the higher the internal and external diameters of the external magnets, the higher the nonlinearity and the difference between maximum and minimum values of the stiffness coefficient curve are (consistent with [22]). When the external magnet has a higher diameter than the internal magnet, the direction of the curve changes, the stiffness coefficient has negative parts, but the movement of the magnet is in the positive part range.

In Figure 7a, it can be seen that the higher the dimensions of the double-sided magnetic spring, the higher the stiffness coefficient is. Comparing it to Figure 7b, it can be seen that the stiffness coefficient, in general, is lower when the external magnet is ring shaped but it also has higher nonlinearity and the difference between maximum and minimum value of the stiffness coefficient is higher for ring-shaped external magnets.

To change the stiffness of the spring, it is better to use external ring-shaped magnets and change their internal diameter, because it changes, only slightly, the shape of the stiffness coefficient curves (only its value). It is also possible to change the height of the external magnets and the air gap between magnets, but too high or too low an air gap causes bending in the curve.

## 4. Induced Voltage

The system consisting of two fixed magnets and one moveable magnet can be represented as mass $m_1$ between two springs with stiffness coefficients $k_{h1}$ and $k_{h2}$ and dampers $b_{h1}$ and $b_{h1}$. The whole system moves due to external vibration and that movement is denoted by $z_0$. That movement will be simulated as a sinus function with amplitude equal to 0.15 mm, which is the averaged amplitude of the real object—the vibration generator supplied with sinusoidal current with amplitude equals 1.2 A. Damping force $F_T$ is acting on the magnet from the voltage induced in coils. The model of that system is shown in Figure 8.

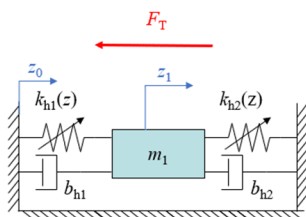

**Figure 8.** Model of the energy harvester.

The forces acting on the moveable magnet can be described using the Lagrange Equation, as follows:

$$\frac{d}{dt}\frac{\partial E_k}{\partial \dot{z}} - \frac{\partial E_k}{\partial z} + \frac{\partial E_p}{\partial z} = F_T - \frac{\partial D}{\partial \dot{z}}, \tag{5}$$

where $E_k$—kinetic energy, $z_0$—known value of movement of the object on which the harvester is, $z_1$—movement of the internal magnet, $E_p$—potential energy, $D$—dissipation, $F_T$—external force acting on the magnet; in this case, that is damping force of the induced voltage.

The kinetic energy is from the magnet movement of the mass of the magnet and is transformed into potential energy. Movement $z_0$ is treated as a known value, so in the equations, only $z_1$ is considered as a variable. Kinetic energy, in this case, does not depend on the displacement, so the partial differential of the kinetic energy in the displacement equals 0. The potential energy is stored in the springs during movement of the magnet. The double-sided magnetic spring consists of the two identical magnetic springs. The calculated stiffness coefficient for the double-sided magnetic spring should be divided by 2 to obtain the stiffness coefficient of a magnetic spring $k_{h1}(z) = k_{h2}(z)$. The variable $z$ is a relative movement of the magnet according to the base, which equals $z_1 - z_0$. The energy is then released to the dampers and dissipates. The damping coefficients $b_{h1}$ and $b_{h2}$ equal 0.0225 and were obtained from the optimization, comparing to the movement of the real object—reference magnetic spring. The equation derived from the Lagrange Equation is similar to the equation of the inertial generator [5] and magnetic springs [12,20,22]. Due to canceling each other out, the gravitation force and static spring force (magnetic force) do not occur in these equations. Taking into consideration the above-mentioned remarks and rearranged Equation (5), the mathematical model, in terms of set of ordinary differential equations representing the energy harvester, OE model, is as follows:

$$m_1\ddot{z}_1 + k_{h2}(z)(z_1 - z_0) + k_{h1}(z)(z_1 - z_0) = -F_T - b_{h2}(\dot{z}_1 - \dot{z}_0) - b_{h1}(\dot{z}_1 - \dot{z}_0). \tag{6}$$

In order to take the external force acting on the magnetic spring into consideration (the acceleration of the external vibration) and to use the relative movement of the magnet as a variable, where $z_1$ equals $z + z_0$, Equation (6) can be described as follows [26]:

$$m_1\left(\ddot{z} + \ddot{z}_0\right) + k_{h2}(z)z + k_{h1}(z)z = -F_T - b_{h2}\dot{z} - b_{h1}\dot{z} - m_1 g. \tag{7}$$

The force $F_T$ is the transducer force from the voltage induced in coils during movement of the magnet, which can be treated as the electrical damping force. This force is calculated from the power, velocity of the magnet, and resistance. Maximum power is obtained when the harvester is loaded with a load with resistance equal to the resistance of two coils [5,28] and is expressed by the following formula:

$$F_T = \frac{e^2 R_L}{(2R_c + R_L)^2 \dot{z}} = \frac{e^2}{8R_c \dot{z}} = \frac{(2NBl\dot{z})^2}{8R_c \dot{z}} = \frac{(2NBl)^2}{8R_c}\dot{z}. \tag{8}$$

where $R_L$—load resistance, $R_C$—resistance of one coil, $e$—induced voltage, $\dot{z}_1$—velocity of the internal magnet, $N$—number of turns of one coil, $B$—magnetic flux density, $l$—length of one turn of the coil.

The induced voltage and the maximum possible to harvest power were calculated for the resonance frequency of each magnetic spring. The resonance frequency was obtained from the square root of the stiffness coefficient $k$ of the double-sided magnetic spring, divided by the mass of the internal magnet $m_1$. For simplicity, it was assumed that the resonance frequency was calculated for the stiffness coefficient that equals coefficient $a_9$, which corresponds to the center of the stiffness coefficient curve. The mass of the internal magnet depends on its dimensions—volume $V_m$ and density—mass per volume $\rho_m$ ($7.5 \times 10^3$ kg/m$^3$). Mass of the middle magnet equals density multiplied by volume. Resonance frequency was, therefore, calculated from Equation (9).

$$f = \frac{\sqrt{\frac{k}{m_1}}}{2\pi} = \frac{\sqrt{\frac{a_9}{V_m \rho_m}}}{2\pi}. \tag{9}$$

The resonance frequency and the mass of the internal magnet are given in Table 5.

**Table 5.** The resonance frequency and the mass of the internal magnet for each case.

| s | c | Resonance Frequency, Hz | Internal Magnet Mass, kg | Acceleration, m/s$^2$ | R, Ω |
|---|---|---|---|---|---|
| Ref | Ref | 117 | 1.77 | 81.25 | 10.32 |
| 1 | 2 | 49 | 3.54 | 14.35 | 20.64 |
| 1 | 3 | 24 | 5.31 | 3.38 | 30.96 |
| 2 | 2 | 190 | 1.77 | 212.95 | 10.32 |
| 2 | 3 | 138 | 5.31 | 112.9 | 10.32 |
| 3 | 2 | 55 | 3.54 | 17.87 | 20.64 |
| 3 | 3 | 29 | 5.31 | 4.88 | 30.96 |
| 4 | 2 | 68 | 7.08 | 27.49 | 17.2 |
| 4 | 3 | 31 | 15.93 | 5.66 | 24.08 |
| ref0 | ref0 | 93 | 1.77 | 51.59 | 10.32 |
| 5 | 2 | 115 | 1.77 | 78.06 | 10.32 |
| 5 | 3 | 107 | 1.77 | 68.26 | 10.32 |

**Table 5.** *Cont.*

| s | c | Resonance Frequency, Hz | Internal Magnet Mass, kg | Acceleration, m/s² | R, Ω |
|---|---|---|---|---|---|
| 6 | 2 | 143 | 1.77 | 121.43 | 10.32 |
| 6 | 3 | 43 | 1.77 | 10.93 | 13.76 |
| 7 | 2 | 59 | 14.16 | 20.32 | 34.4 |
| 7 | 3 | 39 | 47.79 | 9.21 | 72.24 |
| 8 | 2 | 46 | 14.16 | 12.46 | 34.4 |
| 8 | 3 | 31 | 47.79 | 5.71 | 72.24 |

In Figures 9–11, the resultant set of voltage waveforms generated in the winding of the energy harvester, in different variants, is shown. The voltage was filtered by the lowpass Butterworth filter of the 10th order, with the cutoff frequency equal to 1000 Hz and the sample frequency was 5000 Hz.

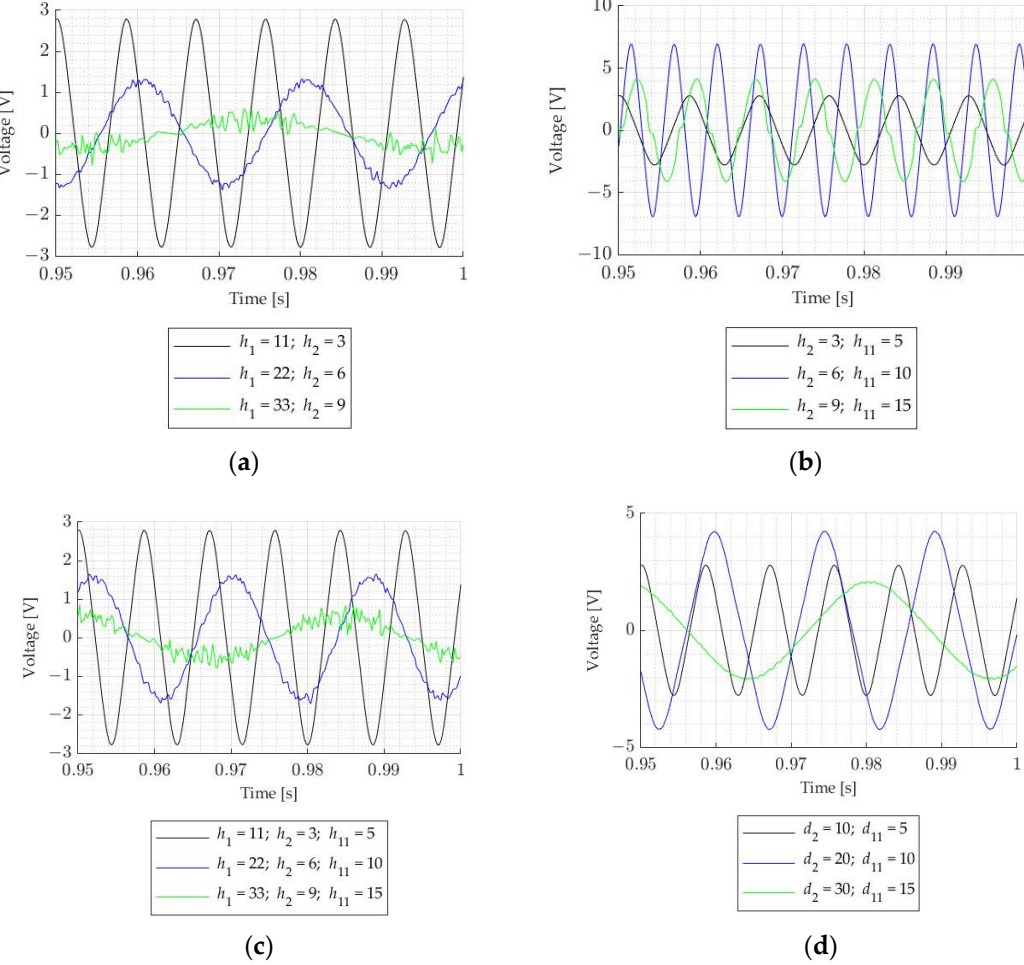

**Figure 9.** The voltage induced in the energy harvester for the resonance frequency of the double-sided magnetic spring: (**a**) set $s_1$ $h_2/h_1$ = const. (**b**) set $s_2$ $h_{11}/h_2 = h_{12}/h_2$ = const. (**c**) set $s_3$ $h_2/h_1$ = const. and $h_{11}/h_2 = h_{12}/h_2$ = const. (**d**) set $s_4$ $d_{11}/d_2 = d_{12}/d_2$ = const.

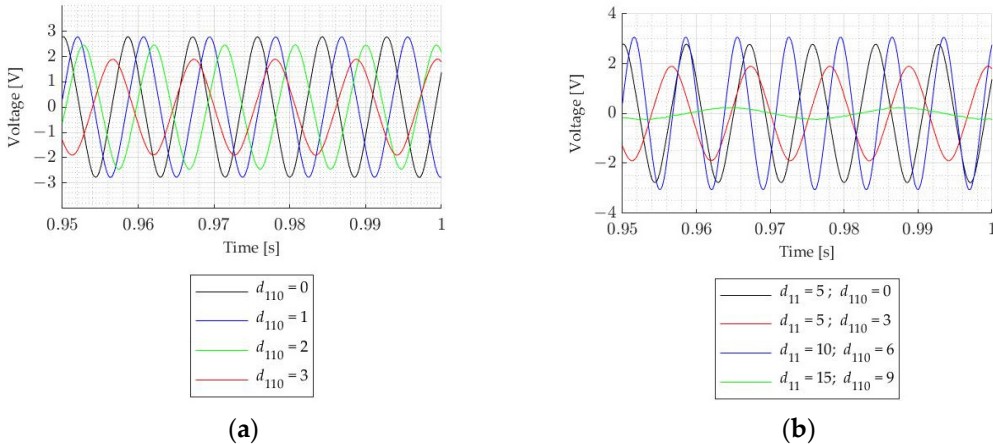

**Figure 10.** The voltage induced in the energy harvester for the resonance frequency of the double-sided magnetic spring: (**a**) set $s_5$ various internal diameters of the external magnets, (**b**) set $s_6$ $d_{110}/d_{11}$ = $d_{120}/d_{12}$ = const.

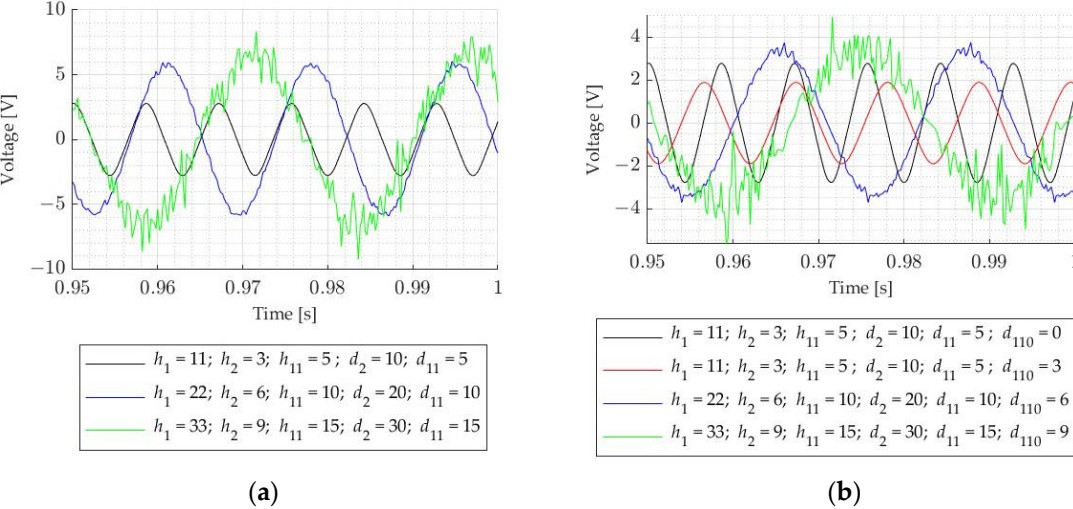

**Figure 11.** The voltage induced in the energy harvester for the resonance frequency of the double-sided magnetic spring: (**a**) set $s_7$ with cylindrical magnets, where all proportions are kept constant, (**b**) set $s_8$ with ring-shaped magnets, where all proportions are kept constant.

Comparing Figure 9a–c, the induced voltage dependencies on the magnet height and the air gap distance can be seen. The higher the air gap and internal magnet, the lower the voltage is. This can be caused by the higher non-linearity in the stiffness coefficient curve. It can be seen, especially in Figure 9b, where higher voltage is induced for the internal magnet 6 mm height and external magnets 10 mm height, but for lower and higher magnets, the induced voltage is lower. For that case, the curve for the stiffness coefficient was also the most linear. The same can be seen in Figure 9d, where for the internal magnet with 20 mm width and the external magnets with 10 mm width, the induced voltage has the highest value; for that case, the stiffness coefficient in the range ±1 mm is nearly linear and the change in that coefficient is lower than for the case with magnets with higher diameters. Comparing Figure 9a,c, it can be seen that for the higher external magnet, the voltage is higher. Considering the voltage in Figure 9b, it cannot be said that the higher the stiffness, the higher the voltage. Therefore, we conclude that, for lower dimensions of the magnet and when magnets are further from each other, their influence on each other is lower, so the magnetic flux density and the induced voltage are lower.

In Figure 10a, it can be seen that the higher the internal diameter of the ring-shaped external magnet, the lower the induced voltage. The differences between voltages are not so high as to not consider benefits from such a solution of changing the stiffness coefficient. In Figure 10b, the same situation as in Figure 9d for the ring-shaped magnet with external diameter 10 mm and internal diameter 6 mm can be seen. The induced voltage is the highest, and for such a configuration, the force and stiffness coefficient are also the highest.

In Figure 11a,b, it can be seen that the higher dimensions in the double-sided magnetic spring, the higher the induced voltage is. For the double-sided magnetic spring with the ring-shaped magnet, the maximum induced voltage is slightly lower (Figure 11b). The higher the dimensions of the magnets, the noisier the waveforms are.

The simulation for the average power generated by the energy harvester with the double-sided magnetic spring for each case and the power regarding the volume of the energy harvester (density of the power) is shown in Table 6. In order to compare each case of the magnetic spring for its resonance frequency power, this could be normalized by division by acceleration [29]. Average power was obtained from Equation (10).

$$P = \frac{1}{t_1} \int_0^{t_1} p(t)dt = \frac{1}{t_1} \int_0^{t_1} \frac{e(t)^2}{8R_c} dt, \tag{10}$$

where $P$—average power, $t_1$—time of the simulation, which was 12 s (with sample time 200e-6 of the real data from laser measurements), $p(t)$—instantaneous power, $e(t)$—instantaneous voltage, $R_c$—resistance of the coil.

**Table 6.** The average simulated power generated by the energy harvester for each case.

| $s$ | $c$ | Average Power, mW | Power Density, mW/m$^3$ | Normalized Power, mW/(m$^2$/s) | Normalized Power Density, mW/(m$^3$m$^2$/s) |
|---|---|---|---|---|---|
| *Ref* | *Ref* | 43.28 | 26.778 | 0.53 | 329.57 |
| 1 | 2 | 4.86 | 1.599 | 0.34 | 111.42 |
| 1 | 3 | 0.38 | 0.085 | 0.11 | 25.19 |
| 2 | 2 | 271.71 | 132.652 | 1.28 | 622.91 |
| 2 | 3 | 94.78 | 38.214 | 0.84 | 338.46 |
| 3 | 2 | 7.59 | 2.349 | 0.42 | 131.45 |
| 3 | 3 | 0.78 | 0.161 | 0.16 | 33.07 |
| 4 | 2 | 57.53 | 15.541 | 2.09 | 565.27 |
| 4 | 3 | 10.31 | 1.55 | 1.82 | 329.57 |
| *ref0* | *ref0* | 20.55 | 13.295 | 0.4 | 257.68 |
| 5 | 2 | 43 | 26.736 | 0.55 | 342.51 |
| 5 | 3 | 33.92 | 21.4 | 0.5 | 313.49 |
| 6 | 2 | 54.04 | 28.107 | 0.45 | 231.48 |
| 6 | 3 | 0.2 | 0.067 | 0.02 | 6.13 |
| 7 | 2 | 59.38 | 8.02 | 2.92 | 394.63 |
| 7 | 3 | 35.51 | 1.78 | 3.86 | 193.33 |
| 8 | 2 | 21.9 | 3.203 | 1.76 | 257.08 |
| 8 | 3 | 10.45 | 0.579 | 1.83 | 101.33 |

From Tables 5 and 6, it can be seen that for the higher frequency and the higher acceleration, higher power is generated. That is why a comparison will be made only by considering the normalized power and the normalized power density. The highest normalized power is generated by the energy harvester with the double-sided magnetic spring with the highest dimensions and cylindrical magnets. However, the normalized power density is the highest for the magnetic spring with 6 mm height internal magnet and 10 mm height external magnets ($s_2c_2$). Slightly lower normalized power density is seen for the magnetic spring with 10 mm width internal magnet and 10 mm width external magnets ($s_4c_2$). Cases $s_6c_3$ and $s_7c_3$ have comparable resonance frequency but the normalized power

density is lower for $s_6c_3$. It can be seen that changing the external ring-shaped magnet's diameter significantly changes the power and that is why it should not be used to adjust frequency. Cases $s_4c_3$ and $s_8c_3$ have the same frequency but the normalized power density is higher for $s_4c_3$; however, the normalized power is nearly the same for both cases. Cases $s_3c_3$ and $s_1c_3$ have resonance frequency comparable to $s_4c_3$ and $s_8c_3$ but the normalized power is ten-times lower than for those cases. It can be seen that changing the height of the internal magnets is not an efficient way to change resonance frequency. In order to slightly change resonance frequency, the internal dimension of the external ring-shaped magnets can be changed (comparing *ref*, *ref0*, $s_5c_2$ and $s_5c_3$), because the normalized power density and the normalized power are comparable. Changing the proportions between the air gap and the internal magnet height seems to be the most correct and easiest way to adjust resonance frequency, based on the normalized power from $s_2c_2$. However, due to limitations in the internal magnet movement and due to lowering the generated power, it can only be used to a certain height in the air gap ($s_2c_3$).

## 5. Conclusions

By changing the parameters for the magnets in the double-sided magnetic spring, the repulsive force and the stiffness coefficient can be altered, but this also alters the energy, which can be harvested.

For the longer distance between magnets and the higher diameters of the magnets, the force is more non-linear. For the higher air gap, the force is lower. For the higher value of internal diameter in the ring-shaped external magnets, the force is lower. The force direction changes for higher magnet diameters. In general, the higher all dimensions of the magnetic spring are, the higher the force and the higher the slope of the linear part of the curve are.

The higher the distance between external magnets, the smaller the stiffness coefficient is. If the internal magnet is too close to the external magnets, the curve is similar to a negative parabolic curve. For the higher diameters of the magnets, the minimum value of the stiffness coefficient curve decreases, and the stiffness curve is a negative parabolic curve. The higher the internal diameter of the external ring-shaped magnets, the lower the stiffness coefficient. When the external magnet has a higher diameter than the internal magnet, the direction of the curve changes and the stiffness coefficient has negative parts. In general, when all dimensions in the double-sided magnetic spring are higher, the stiffness coefficient is higher. The stiffness coefficient, in general, is lower when the external magnet is ring shaped but it also has higher non-linearity.

To change the stiffness in the spring without changing the stiffness curve shapes, external ring-shaped magnets with different internal diameters can be used.

To adjust the resonance frequency of the external and internal diameters of the external magnets, the height of the magnets should not be used. To adjust the resonance frequency only slightly (by a few Hz), the internal dimension of the external ring-shaped magnets can be changed. It seems that changing the proportions between the air gap and the internal magnet height is the best way to adjust the resonance frequency.

The next step in improving the energy harvester and increasing the generated power is to determine the best placement and parameters for the coil.

**Author Contributions:** Conceptualization, J.B., T.T. and M.S.; methodology T.T.; mathematical modelling J.B.; computation, data curation and investigation T.T. and J.B.; formal analysis, T.T.; resources, M.S.; writing—original draft preparation, J.B., T.T. and M.S.; writing—review and editing J.B., T.T. and M.S.; supervision, T.T. All authors have read and agreed to the published version of the manuscript.

**Funding:** This research received no external funding.

**Data Availability Statement:** Data is contained within article.

**Conflicts of Interest:** The authors declare no conflict of interest.

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
