# Peer review of "Simulation and Investigation of the Change of Geometric Parameters on Voltage Induced in the Energy Harvesting System with Magnetic Spring"

_electronics, doi:10.3390/electronics11101639_

Round 1
Reviewer 1 Report
The literature review section is missing a significant part of reviewing previous studies and advances in the field. The introductions should clearly address the problem or shortcoming in the field which this effort aims to address, improve or provide a solution. The author should clearly express what is the contribution of this research to the-state-of-art and what is the gap in the field that they are addressing.
Otherwise, the magnetic springs are the most common mechanisms used to build vibration energy harvesting systems and several industrial versions are already developed and available. Mathematical models, vibration equations, and nonlinearity analysis equations of these types of harvesters are well developed, available, and publicized.
In section 2: Author selected several magnet shapes and combinations without explaining the reasons, selection process or method.
In section 3: Author makes several claims and assumptions in driving mathematical equations without citing required references or providing enough reasoning.
The results of this study are solely based on simulation data and there is no experimental verification or comparison with other verified data.
The paper in the current form resembles a report from several simulation work without verification. It lacks required scientific support in problem definition and result verification.
The dimensional optimization of magnetic energy harvesters for specific applications have already been done and the process and methods are well established.
Author Response
Thank You for Your contribution to the review of our article. We are very grateful for valuable comments and suggestions, following them allowed us to improve the transparency and quality of our article. Below we have prepared answers to Your comments:
We have improved our introduction, by adding some information about recent works and about our goal which was: to find the easiest and most efficient way (without significant drop of the harvested energy) of adjusting resonance frequency while the energy harvester is working.
Shapes of the magnets and combinations of the magnetic spring were chosen based on the prototype of the magnetic spring on which another investigation was conducted. We want also to find dependencies between some of the dimensions. That is why we chose to change them proportionally and simultanously. The dimensions were two and three times higher than the prototype dimensions (for the reference spring). That will allow us to conduct measurements in the future works.
The equations were improved and supported by citations. Some of the assumptions were made based on the previous investigation of the magnetic spring. Movement of the base is the averaged movement of the vibration generator (it will allow us to conduct measurements in the future works). The damping coefficient was calculated by the optimization after the measurement of the magnet movement and compared to the model of the magnetic spring.
Some comparisons to existing data were added:
- Tri Nguyen, D.A. Genov, H. Bardaweel, Vibration energy harvesting using magnetic spring based nonlinear oscillators: Design strategies and insights, Applied Energy, 269 (2020) 115102
- Li, S. Wu, P. C. K. Luk, M. Gu and Z. Jiao, "Enhanced Bandwidth Nonlinear Resonance Electromagnetic Human Motion Energy Harvester Using Magnetic Springs and Ferrofluid," in IEEE/ASME Transactions on Mechatronics, vol. 24, no. 2, pp. 710-717, April 2019, doi: 10.1109/TMECH.2019.2898405.
Reviewer 2 Report
The paper entitled "Simulation and investigation of the change of geometric parameters on voltage induced in the energy harvesting system with magnetic spring" models and simulate the voltage induced during the movement of a moveable magnet in the double-sided magnetic spring. The aim is to control the stiffness of the double-sided magnetic spring and allow changes in the natural frequency of the energy harvester system
The manuscript needs proofreading. Logic and grammatic errors must be corrected. Figures are a little blurry, and the font series used in the text area is not the same.
The references are not adequate. Only 5 of 14 references are from the last 5-years. Is this a research scientific area of interest?
Also, the authors should carefully choose the plots to be presented to help the reader comprehend the thinking flow.
The authors did not present any comparison with other works. Is this subject new? Are other scientists interested and working in the field? Is this work significant and penetrating the scientific community?
Author Response
Thank You for Your contribution to the review of our article. We are very grateful for valuable comments and suggestions, following them allowed us to improve the transparency and quality of our article. Below we have prepared answers to Your comments:
Some errors and mistakes are corrected. The quality of pictures has been improved and colors are now more visible.
We have improved our introduction, by adding some information about recent works and about our goal which was: to find the easiest and most efficient way (without significant drop of the harvested energy) of adjusting resonance frequency while the energy harvester is working.
Plots and combinations of the magnetic spring were chosen based on the prototype of the magnetic spring on which another investigation was conducted. We want also to find dependencies between some of the dimensions. That is why we chose to change them proportionally and simultanously. The dimensions were two and three times higher than the prototype dimensions (for the reference spring). That will allow us to conduct measurements in the future works.
Some comparisons to existing data were added:
- Tri Nguyen, D.A. Genov, H. Bardaweel, Vibration energy harvesting using magnetic spring based nonlinear oscillators: Design strategies and insights, Applied Energy, 269 (2020) 115102
- Li, S. Wu, P. C. K. Luk, M. Gu and Z. Jiao, "Enhanced Bandwidth Nonlinear Resonance Electromagnetic Human Motion Energy Harvester Using Magnetic Springs and Ferrofluid," in IEEE/ASME Transactions on Mechatronics, vol. 24, no. 2, pp. 710-717, April 2019, doi: 10.1109/TMECH.2019.2898405.
Reviewer 3 Report
In this paper, the parameter influence of energy harvester with magnetic spring is deeply studied, which provides an effective reference for the design of related fields.
1. Lack of introduction to the content of the article in the last paragraph of the Introduction.
2. The quality of the pictures in the full text is very poor, and the curves in the picture are difficult to see clearly.
3. For the energy harvester, the excitation conditions are important. The research done in this paper is under what kind of incentives and whether the incentive conditions have any influence on the choice of parameters.
4. Please summarize several guiding conclusions and list them by article.
5. What is the significance of the s3 experiment? Are the parameter settings of this group of experiments wrong?
Lines 6.216-217: The expression "Lowering distance between...the curve" has no corresponding experimental support
7. Lines 167-168: Figure 3 c) is wrong, it should be Figure 2c); and the conclusion of this sentence is that the distance between the magnets becomes larger, the magnetic force enhancement is not in line with the conventional, and the conclusion is inconsistent with the experiment.
Line 8.105: "The dimensions that were..." is unclear
9.Table1: The third line of s2c2 has an error in the h11 subscript; Table5: The conclusions of s2c2 and lines 332-333 are inconsistent
Author Response
Thank You for Your contribution to the review of our article. We are very grateful for valuable comments and suggestions, following them allowed us to improve the transparency and quality of our article. Below we have prepared answers to Your comments:
- We have improved our introduction, by adding some information about recent works and by adding last paragraph in which we have described content of each section.
- The quality of pictures has been improved and colors are now more visible.
- Some of the assumptions were made based on the previous investigation of the magnetic spring. We have investigated moveable magnet movement while the magnetic spring was placed on the vibration generator, in order to improve model of the magnetic spring. The movement of the base is the averaged movement of the vibration generator (it will allow us to conduct measurements in the future works). That is why shapes of the magnets and combinations of the magnetic spring were chosen based on the prototype of the magnetic spring on which another investigation was conducted. The dimensions were two and three times higher than the prototype dimensions (for the reference spring). That will allow us to conduct measurements in the future works.
- The conclusions have been summarized and listed by the article in the conclusion chapter.
- The third case of the magnetic spring was added to check which of two parameters have the most impact on the change of the stiffness.
- The expression "Lowering distance between...the curve" is corrected to “Lowering gap between magnets by changing internal magnet dimensions without changing the distance between external magnets causes linearization of the curve, but if the internal magnet is too close to external magnets, the curve is similar to the negative parabolic curve.”
- The conclusion is now corrected to: “In Figure 2 c) it can be seen that the higher the magnets are and the longer distance between them is, the higher the maximum force acting on the internal magnet is. However; in the internal magnet movement range the force is lower for higher magnets and the distance between them.”
- The expression "The dimensions that were..." is corrected to: „Dimensions were altered to maintain constant proportions between some of them. The dimensions that were altered are shown in Figure 1 a).”
- Subscripts are corrected in table 1. The table 5 is now table 6 and the whole conclusions are altered. Some of the parameters like frequency and acceleration are added in table 5 and in conclusions the normalized power density and the normalized power for all cases are compared.
Reviewer 4 Report
This manuscript studies the mathematical modeling and simulation of the voltage induced during the movement of a moveable magnet. Here are some comments for the authors.
1 The introduction needs to be rewritten. In the introduction, the background needs to be introduced first, then the current developments need to be comprehensively summarized, and some research deficiencies are also explained. Finally, the work to be done in this paper and the innovation points are introduced. The author can refer to the introduction writing style of these two papers.
[1] H. Tri Nguyen, D.A. Genov, H. Bardaweel, Vibration energy harvesting using magnetic spring based nonlinear oscillators: Design strategies and insights, Applied Energy, 269 (2020) 115102.
[2] D. Zou, G. Liu, Z. Rao, T. Tan, W. Zhang, W.H. Liao, Design of a multi-stable piezoelectric energy harvester with programmable equilibrium point configurations, Applied Energy, 302 (2021) 117585.
2 The references in the introduction are not comprehensive enough, and there have been many studies on energy harvesting by magnets and magnetic springs. The author's explanation of this field is not comprehensive enough.
3 The quality of the figures in the paper needs to be improved, and many lines and colors can't be seen clearly.
4 In fig. 5, fig. 6, and fig. 7, when the displacement of the middle magnet is greater than some value, the stiffness decreases instead, which means that the system is unstable from the dynamic point of view. The author needs to explain the reasons. Is it because the repulsion force between magnets becomes the attraction force?
5 There are some errors in Equation 5. (a) z0 is the movement of the base, so why does it only affect Kh1 and bh1, but not Kh2 and bh2? (b) Gravity counteracts the static force in the magnetic force, so it should not appear in Equation (3). Because in the application, the vibration acceleration of the base is usually known, not the vibration displacement and vibration speed. Therefore, Equation 5 needs to be changed to take the acceleration of the base as a variable, instead of the present form (Such as Eq. 23 in the following reference).
[1] H. Tri Nguyen, D.A. Genov, H. Bardaweel, Vibration energy harvesting using magnetic spring based nonlinear oscillators: Design strategies and insights, Applied Energy, 269 (2020) 115102
6 There is an error in Equation 6. In the damping force, the vibration speed appears in the denominator?
7 In Figure 9 to Figure 11, what is the resonance frequency of each energy harvester? There are still many parameters that must be given, such as magnet mass, residual magnetic field strength, external resistance load, excitation acceleration level, etc.
8 The energy harvester focuses on the harvesting power per unit volume. The larger the volume, the greater the power obtained, but the power density is not necessarily high. Therefore, the harvesting power in Table 5 needs to be normalized, such as Eq. 25 in the following reference
- Zou, K. Chen, Z. Rao, J. Cao, W.-H. Liao, Design of a quad-stable piezoelectric energy harvester capable of programming the coordinates of equilibrium points, Nonlinear Dynamics, (2022).
9 How to get the average power?
10 The conclusion needs to be simplified. Just give the most important findings.
11 Authors need to learn how to write papers, and there are many mistakes in them.
Author Response
Thank You for Your contribution to the review of our article. We are very grateful for valuable comments and suggestions, following them allowed us to improve the transparency and quality of our article. Below we have prepared answers to Your comments:
- We have rewritten our introduction according to Your suggestions. We have also added last paragraph in which we have described the content of each section.
2.We have improved introduction by adding some information about recent works and about our goa which was: to find the easiest and most efficient way (without significant drop of the harvested energy) of adjusting resonance frequency while the energy harvester is working.
- The quality of pictures has been improved and colors are now more visible.
- In the article the explanation for changing direction of the stiffness has been added. „When the internal magnet is near the one of external magnets, the influence of that external magnet is higher than the influence of the second external magnet. The closer magnets are to each other the more significant the attraction force becomes. The lower is the height of the middle magnet, the sooner it happens, it means for higher distance between middle and external magnets. The middle magnet will vibrate or it will rotate if it doesn't have any guiding rod.”
- The equations have been improved and supported by citations. The z0 is added to influence kh2 and bh2. Some of the assumptions were made based on the previous investigation of the magnetic spring. The movement of the base is the averaged movement of the vibration generator (it will allow us to conduct measurements in the future works). We have changed equation 5, according to Your suggestion, to take acceleration of the base instead of vibrations..
- The equation 7 (which was equation 6 before) is improved in order to show dependencies. The damping force is actually a transducer force, which is calculated from the power and velocity of the magnet. This explanation is now added to the article.
- The resonance frequency was calculated from the coefficient a9 and mass of the magnet. The density of the magnet equals 7.5×103 kg/m3 and the mass of the magnet depends on its volume as density multiply by the volume. The parameters of the magnet were taken from the side of the magnet manufacturer https://enesmagnets.pl/shop/en/permanent-magnets/neodymium-sintered/cylindrical-magnets/d5-x-5-n38-ndfeb-neodymium-magnet.html.That information is also added to the article. The table in which the resonance frequency, acceleration and magnet mass are given, was added.
- The table with the generated power is changed to table 6, in which not only the generated power but also the normalized generated power and the normalized generated power density are added.
- The average power was obtained from the waveform, from the integral of instantenous power and from the time of the simulation which was 12 s. It is time of the measurement of the magnet movement in magnetic spring on the vibration generator. The sample time was 200e-6, it was the sample time for laser measurement. These measurements were done during another investigation of the magnetic spring.
- The conclusions are now simplified and listed by the article in the conclusion chapter.
Round 2
Reviewer 3 Report
The authors have made reasonable revisions to the paper and the content of the manuscript now meets the requirements for publication.
Reviewer 4 Report
Thank the authors for all the revisions. I think the manuscript can be acceptable in present form.